

# Vertical distribution of halogenated trace gases in the summer Arctic stratosphere determined by two independent *in situ* methods

Johannes C. Laube[1,2], Tanja J. Schuck[3], Huilin Chen[4], Markus Geldenhuys[5], Steven van Heuven[4], Timo Keber[3], Maria Elena Popa[6], Elinor Tuffnell[2], Bärbel Vogel[1], Thomas Wagenhäuser[3], Alessandro Zanchetta[4], and Andreas Engel[3]

[1]Institute of Climate and Energy Systems (ICE-4: Stratosphere), Forschungszentrum Jülich, Jülich, 52425, Germany
[2]School of Environmental Sciences, University of East Anglia, Norwich, NR4 7TJ, United Kingdom
[3]Institute for Atmospheric and Environmental Sciences, University of Frankfurt, Frankfurt (Main), 60438, Germany
[4]Centre for Isotope Research, University of Groningen, The Netherlands
[5]South African Weather Service, 1263 Heuwel Road, Centurion, South Africa
[6]Institute for Marine and Atmospheric Research Utrecht, Utrecht University, The Netherlands

*Correspondence to*: Johannes C. Laube (j.laube@fz-juelich.de)

**Abstract.** Many halogenated trace gases are important greenhouse gases and/or contribute to stratospheric ozone depletion, yet their spatial distribution and temporal evolution in the stratosphere remain poorly constrained. We here present a new high-altitude dataset of a large range of these gases. The results are based on a large balloon flight in the Arctic in summer 2021. Air samples were collected using a passive (AirCore) as well as an active (cryogenic) technique; the former being the largest AirCore flown to date, thus enabling the quantification of an expanded variety of halogenated gases. The evaluation of the results demonstrates good comparability in most cases, but also revealed strengths and weaknesses for both sampler types. In addition, we show examples of the scientific value of this data, including the identification of air masses likely originating from the Asian Monsoon region, and the derivation of the average stratospheric transit times (i.e., the mean ages of air) from multiple tracers.

## 1 Introduction

Due to the ongoing effects of their decomposition products on the stratospheric ozone layer, the quantification of Ozone Depleting Substances (ODSs) remains important. This has recently been reemphasised as multiple ODSs have been found to either not decrease in global abundances as expected, or even increase (Vollmer et al., 2021; WMO, 2022; Western et al., 2022, 2023). In addition, many of these species – including related non-ODS fluorinated compounds – are strong greenhouse gases and have atmospheric lifetimes on the order of decades to millennia, with emissions therefore creating a long-lasting legacy in the atmosphere (Droste et al., 2020; Simmonds et al., 2020; Stanley et al., 2020; Say et al., 2021). While even the near-ground observations are unable to constrain the regional sources of global emissions for many halogenated species (Weiss et al., 2021), the data sparsity is distinctly more pronounced at the hard-to-reach stratospheric altitudes. Satellite data




can offer good quality, but only for some species over certain altitude ranges (Stiller et al., 2012; Kolonjari et al., 2024; Saunders et al., 2024), with some products exhibiting higher uncertainties (Harrison et al., 2012) or obvious biases (Dodangodage et al., 2021). Here we present a unique dataset based on the established AirCore passive sampling technique

(Karion et al., 2010; Membrive et al., 2017; Engel et al., 2017; Laube et al., 2020; Wagenhäuser et al., 2021; Tans, 2022). In this case, a very large AirCore was flown alongside a cryogenic whole-air sampler, which allows for an independent verification of the AirCore method; something that has not been performed for halogenated species before. A similar verification for the major anthropogenic greenhouse gases $CO_2$, $CH_4$, $N_2O$, and $SF_6$ can be found in our companion paper, i.e., Schuck et al. (2024). In addition, due to the increased volume of the stratospheric and upper tropospheric air that was

retrieved with this AirCore, a much-improved number of halogenated trace species (as compared to the previously published seven in Laube et al., 2020, and Li et al., 2023) could be quantified (including CFCs, halons, HCFCs, HFCs, PFCs, and chlorocarbons), alongside retaining a relatively good altitude resolution. Describing the details of the employed methods and evaluating and comparing the results from both instruments are the primary focus of this manuscript.

## 2 Methods

The results presented below are based on the flight of a gondola weighing approximately 345 kg, which was lifted to ~32 km by a large balloon launched by the French Space Agency CNES on 12[th] August 2021 from the Esrange base near Kiruna, Sweden (67.8883 °N, 21.0847 °E, 331 m.a.s.l.) as part of the EU infrastructure Hemera. Apart from the two instruments described here the payload contained several more, which focused mainly on measurements of major greenhouse gases and for which results are described elsewhere (Schuck et al., 2024).

### 2.1 Cryosampling

The cryogenic whole-air-sampler has been operated for several decades (largely by the University of Frankfurt, e.g., Schmidt et al., 1987; Engel et al., 2002, 2009) and is one of the most well-established stratospheric air sampling platforms in existence. Its ability to retrieve large air samples from the low-pressure stratosphere has repeatedly proven extremely useful for the investigation of a multitude of trace gases (including, in some cases, their isotopic composition, e.g., Röckmann et al.,

2011); especially at altitudes outside of the reach of aircraft (above ~20 km), with the cryosamplers able to reach up to ~40 km. As it has been described in detail multiple times, we here only present a brief summary of its operating principles: It consists of a Dewar vessel filled with liquid neon and cooling a number of stainless-steel containers to around -240 °C. This setup acts as a cryopump when one of the containers is opened by telecommand during the flight, thus allowing the collection of several litres of air over well-defined and narrow altitude ranges. Sampling is concluded by mechanically

crushing a gold tube that is part of the inlet and therefore cold-welding the container shut. More details on this particular flight can be found in a companion paper (Schuck et al., 2024). Most importantly, the inlets of five containers were equipped with cotton filters to catalytically prevent the sampling of ozone. From the available total of 15 canisters, 13 samples could



be successfully collected, four of which had a catalyst. The total weight of the cryosampler excluding the (required) gondola is around 60 kg.

## 2.2 AirCore technique and MegaAirCore

To introduce AirCores, these were invented at the NOAA Global Monitoring Labs (Karion et al., 2010) and are passive sampling devices based on a long (normally up to ~100 m, except for the "high-resolution" AirCore: 300 m, see Membrive et al., 2017) coil of extremely thin-walled stainless-steel tubing. Due to the low weight these can be launched up to ~35 km using comparably low-cost weather balloons. As the tubing is open to outside air on one end, it empties during ascent, and then collects a vertical profile of air after the balloon has burst, i.e., during its descent on a parachute. As the internal diameter of the tubing is typically very small (3-8 mm), diffusion limits mixing inside the AirCore coil (i.e., over the length of the tube), enabling subsequent trace gas analysis and reconstruction of the vertical mixing ratio distributions provided the payload is retrieved and measured withing a few hours after landing. An alternative is the prevention of mixing by storing segments of AirCore air in so-called subsamplers (Mrozek et al., 2016; Laube et al., 2020), which is essential for more time-consuming measurements, or in the case of balloons being carried out far away from the analytical equipment after the launches. Typical AirCore volumes are between 500 and 1500 ml (exception again Membrive et al., 2017: ~3 litres).

The AirCore used in this study is the so-called "MegaAirCore" (MAC); a definitive exaggeration regarding its name, as it is neither in terms of its length (230 m) nor internal volume (11.1 litres) a million times larger than normal-sized AirCores. The MAC was constructed specifically for this flight and consists largely of two connected pieces of coiled tubing, i.e., 170 m of ¼ inch tubing (inner radius: 5.842 mm, stratospheric end) and 60 m of ½ inch tubing (inner radius: 11.684 mm, tropospheric end), both of which were stainless steel which had been Silco™-1000-coated to provide extra trace gas inertness. Due to the long flight duration of ~5.5 hrs and the cold stratospheric and upper tropospheric temperatures (~-72 °C in the vicinity of the tropopause), the MAC was heated with heating foil (heating power: 130 W/m$^2$, 24V, from Osnatherm GmbH) and using a power bank (capacity: 52800 mAh, from XTPower.com) to between 10 and 13 °C throughout the entire flight. This temperature was monitored with two Arduino data logger (from Adafruit Industries, LLC, type: Feather) containing five temperature sensors each, which were attached to the coils in different locations (Fig. 1), notably including in the vicinity of the inlet which is most exposed to the cold surroundings. The purpose of these coil temperature stabilisation and control procedures was to avoid any loss of the target trace gases – some of which have relatively high boiling points of around 90 °C – to the internal AirCore surfaces. All air entering the MAC was dried using a stainless steel cartridge filled with loosely packed Mg(ClO$_4$)$_2$ as is common for the AirCore technique (Karion et al., 2010, Membrive et al., 2017, Laube et al., 2020). Further temperature stability was provided by placing the coils inside a PE-foam-based package (PLASTAZOTE® low density polyethylene foam LD29, blown with N$_2$ to prevent any target trace gas contamination) as well as covering the outsides of this parcel with aluminium foil (Fig. 1). The mounting of both the MAC and the cryosampler on the well-established TWIN gondola (Schmidt et al., 1987; Engel et al., 2002) is also shown in Fig. 1.



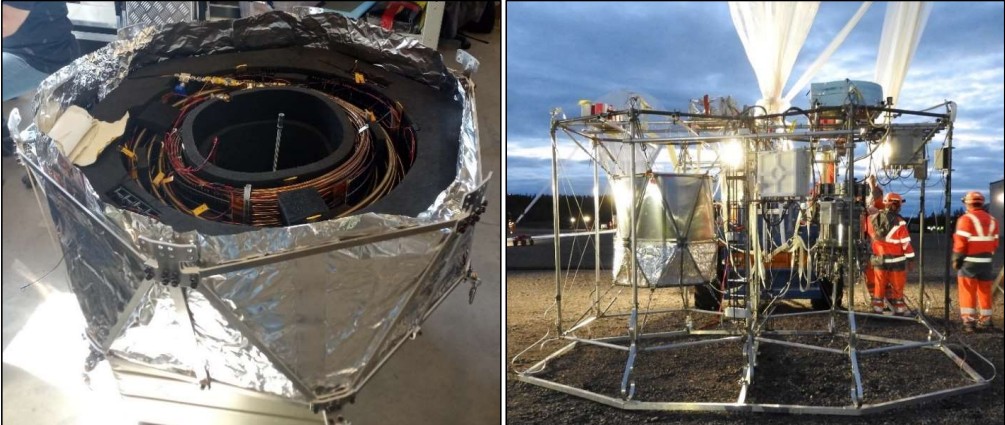

Figure 1: Left: The MegaAirCore (MAC) tubing inside its two layers of insulation. Also visible are several of the temperature sensors deployed to monitor the temperature of the tubing in various places. Right: Gondola just before launch with the MAC parcel mounted low inside the lefthand side hexagon, and the cryosampler inside the righthand side one. Copyright J. C. Laube (left) and CNES/Prodigima - R. Gaboriaud (right).

### 2.3 Subsampling

As halogenated trace gas analysis was not possible on site, the air collected with the MAC needed to be swiftly divided into sections to prevent further mixing, which was achieved using a subsampling technique (Laube et al., 2020). First, shortly after landing the MAC inlet valve was closed manually by the recovery team to prevent extended exchange of the sampled air with near ground outside air. This is less time-critical for the MAC in comparison to other AirCores (Karion et al., 2010; Membrive et al., 2017), as only the stratospheric and upper tropospheric part was being analysed subsequently. The gondola was then transported back to the launch location via helicopter, which caused, when also including the time required for dismounting the MAC from the gondola and preparing it for subsampling, a delay of ~3.5 hrs; the subsampling procedure itself taking ~1.25 hrs. These times are important as mixing induced by diffusion within the MAC was possible prior to subsampling, which affects the achievable altitude resolution. Two exemplary estimates of the root mean square molecular diffusion over the entire period for $SF_6$, and CFC-11 can be found in the supplement (Figure S1 and S2). These are on the order of 1 m in both cases, equivalent to ~27 ml in the most important ¼ inch part of the MAC. Overall, it is worth noting that the gases analysed for the MAC have much smaller diffusion coefficients than $CO_2$ and $CH_4$ (e.g., Table S1 in Martinerie et al., 2009), which helps limiting such mixing effects.



The stratospheric end of the MAC was then connected to a ¼ inch stainless steel cross (Swagelok™) with a vacuum pump (Edwards nXDS10), a high precision pressure sensor (MKS Baratron, range 0-2000 mbar), and a subsampler at the three

other ends. The latter device was almost the same as the Improved Subsampler (ISS) described section S1.1.1 of Laube et al. (2020). It consisted of a central 32-port 1/8-inch valve (from VICI, Switzerland), with a common in- and outlet, as well as loops of ¼ inch stainless steel (~20 ml volume each) attached to each but the first two pairs of ports. The first pair of ports was blanked for a default position that can be exposed to lab air when connecting the AirCore; whereas the second pair was connected to a larger sample (~103 ml) made from connecting two 50 ml stainless canisters with 90°-angled connectors to

form a "loop". This larger sample was installed to collect and store the first bit of air contained in the MAC, which was expected to consist almost entirely of remaining fill gas (here: nearly trace gas-free $N_2$, details in section 3.1). The other 14 loops were intentionally smaller to enhance the attainable altitude resolution, which is otherwise rather limited as the passive sampling techniques yields increasingly smaller amounts of air at the much-decreased pressure at higher altitudes. Each loop of the subsampler had been leak-checked and conditioned via five fill-evacuate cycles with AirCore fill gas prior to the

launch and left filled to ~1 bar. Directly before subsampling each loop was pumped down to < 1mbar in a first cycle, and to < 0.01 mbar in a second one; with a third cycle to ascertain that the vacuum was holding. Then the MAC was opened with the ISS in position 1 (blanked) and the valve to the pressure sensor open (but closed shortly afterwards) to determine the pressure inside the AirCore, which is typically slightly above ambient due to the gradual warming of the tubing after landing (i.e., in the hangar). The tropospheric end of the AirCore was then opened to avoid pressure gradients between the

subsampler loops, followed by the opening of the stratospheric end to each loop successively. The same procedure was repeated for two further subsamplers of the same build (but each having 15 loops @ 103 ml), which were also constructed specifically for this flight. Finally, 6 SilcoCans™ (volume: ~400 ml) were filled from the MAC in the same way, resulting in a total subsampled volume of 5.9 litres, equivalent to 53 % of the full MAC volume (which is why opening the MAC on the tropospheric side for subsampling did not lead to sample contamination with lab air). Also similar to Laube et al. (2020), all

metal subsampler surfaces were Silco-1000-treated to improve surface inertness during sample storage.

**2.4 Analytical techniques for trace gas analysis and quality assurance**

After shipping the samplers back to Germany, dry air mole fractions of halogenated trace gases (referred to here as mixing ratios) were derived using three measurement systems. At Forschungszentrum Jülich (FZJ), all samples (MAC + CRYO)

were processed with an analytical system and methodology equivalent to a previously well-established one (Laube et al., 2010, 2020; Leedham Elvidge et al., 2018; Adcock et al., 2021). This newer system has recently been proven to perform equally well (also in comparison to other internationally recognised measurements) for eight CFCs over a temporal range of several decades at mixing ratios between 0.06 and ~80 ppt (Western et al., 2023). Trace gases are being cryogenically extracted and pre-concentrated using an ethanol/dry ice mixture, then thermally desorbed with freshly boiled water,

separated by gas chromatography (Agilent 6890 GC with a 60 m long GS GasPro column temperature-programmed to heat



from -10 to 200 °C) and detected at high mass resolution (~1000 at 5 % peak height) with a triple-sector mass spectrometer (Waters AutoSpec MS) in selected ion monitoring (SIM) mode. Typical detection limits for the trace gas analysis of a few hundred ml of air are in the lower and sub-ppq (parts per quadrillion) range. The analytical system at Frankfurt University (GUF, CRYO measurements only) is similar in many ways, so we only point out the significant differences here: The

cryogenic extraction and pre-concentration is based on a Stirling cooler and trace gas desorption occurs via resistive heating (Schuck et al., 2018). While a very similar GC (Agilent 7890) and the same column (albeit at only half the length) are being used, the temperature program only starts at 50 °C (preventing the analysis of very low-boiling species such as $SF_6$ or $C_2F_6$), and the MS is a quadrupole-based one (Agilent 5975C). Typical detection limits are in the sub-ppt range with values around 0.1ppt for most substances discussed here. Additionally, all cryogenically collected samples were analysed at GUF for their

content of $SF_6$ and CFC-12 with a well-established system based on gas chromatography with electron capture detection (GC-ECD, Jesswein et al., 2021). In total, 39 halogenated species could be successfully quantified, with 24 measured on two analytical systems. More measurement details, including on average precisions, calibration scales, ions used for quantification, and the level of agreement between labs, can be found in Tables S1 and S2. Additionally, the main working standard (consisting of clean air collected with a trace gas-free metal-bellows compressor in the Taunus mountains near

Frankfurt) used for assigning mixing ratios to samples at GUF was transferred to FZJ and measured alongside the cryosampler, thus enabling a consistent calibration scale for all species reported. A summary of this comparison can be found in the supplement.

## 3 Results and Discussion

### 3.1 Quantification of the fill gas fraction remaining in the AirCore

One of the main challenges of the AirCore technique is the determination of the exact amount and location of fill gas remaining inside the tubing at the start of the descent part of the balloon flight. This fill gas portion mixes with the uppermost air sampled and therefore complicates the determination of trace gas mole fractions as well as the retrospective altitude assignment procedure. Commonly this is being addressed by using a fill gas with a CO mole fraction (1-10 part per million, ppm) that is much higher than those typically found in the middle stratosphere (10's of part per billion, ppb). This

does however create some uncertainty in the air collected at the highest altitudes, which presents a mixture of the two concentrations that can only be constrained with theoretical calculations. To circumvent this problem – and to also have a fill gas fraction quantifiable with the mass spectrometer system at hand – the fill gas employed here was virtually trace gas-free $N_2$. Two exceptions to that trace gas content were i) small (sub- to low part per trillion, ppt) impurities that could be corrected for, as well as ii) ~160 ppt of perfluoro triethylamine (PFTEA, $C_6F_{15}N$), which is not currently present in the

unpolluted atmosphere in detectable amounts (detection limit of the MS system: ~0.01 ppt for a 100 ml sample measured @ $m/z$ 114, i.e., the $C_2F_4N^+$ fragment). A sample of the undiluted fill gas was measured close in time to the MAC samples to ensure appropriate calibration.



In addition, the sensitivity drift of the analytical system over the course of a measurement day (11-16 hrs) needs to be
corrected for. This is typically done by interspersing a "working standard" of air with well-known and close-to-atmospheric
mixing ratios in between sample measurements. However, as the PFTEA is not present in the utilised standard, a substitute
trace species needed to be found which showed a reliable response ratio to PFTEA over time, i.e. within one day as well as,
ideally, over longer periods of weeks and months. Using nine occasions between 2017 and 2023 where varying amounts of
the fill gas (15-100 ml) were measured repeatedly over the course of one day (including at very different times), six
compounds were evaluated for this purpose. H-1211 ($CF_2BrCl$, measured on $m/z$ 129) gave the most consistent ratio between
the signal area of PFTEA in the fill gas and the H-1211 peak area in the surrounding standards interpolated to the time of the
fill gas measurement (average standard deviation of 1.4 %). As our best estimate of the fill gas content uncertainty, we here
use the square root of the sum of squares of i) the aforementioned uncertainty in the response ratio (except for fill gas signals
close to the detection limit, i.e. fill gas fractions less than ~0.3 %: independently determined average precision: 4.6 %), and
ii) the precision of the H-1211 measurements on the respective day.

Figure 2 shows the determined fill gas fractions as a function of the subsampled volume including their uncertainties. As can
be seen, the fill gas fraction drops rapidly to ~0.2 % after ~500 ml. After that, values do not consistently decrease further
instead showing variability between fill gas fractions of 0.3 % and below detection limit of around 0.05 %. This is likely
caused by a small fraction of fill gas remaining in the subsamplers after conditioning (and prior to filling). It does not cause a
problem with the correction as the same fill gas was used for both the MAC and the subsamplers prior to filling. To quantify
how much fill gas might be originating from the subsamplers, we perform two calculations of the total volume of fill gas left
in the MAC: Firstly, via the minimum pressure measured during the balloon flight (11.05 mbar at the highest altitude) and
the total AirCore volume and final pressure after landing, which yields 128.6 ml. Secondly, from simply summing up all fill
gas volumes determined through the PFTEA measurement-based method, which gives, when corrected for the coil
temperature changes from 19 °C during subsampling to on average 11.4 °C during the flight, 132.6 ± 4.5 ml. The two
estimates are not significantly different without even considering the radiosonde pressure uncertainty (<0.1 hPa), with a total
of 4.0 ± 4.5 ml or on average 0.07 % of the total remaining fill gas amount possibly contributed by the remaining subsampler
filling.

Finally, there is one distinct outlier in Figure 2 (circled), i.e., the anomalously high 3.75 % of fill gas determined in the last
loop of the first subsampler. Due to the cyclic nature of the central subsampler valve, this loop is directly adjacent to the
initial position before subsampling, in which the valve remained for ~10 minutes during i) the pressure measurement of the
MAC, and ii) the equilibration to ambient pressure after opening the tropospheric end valve of the MAC. The likeliest
explanation for this outlier is therefore some cross-port leakage during that time. All other loops where however only
exposed to similar pressure gradients for 20 seconds as the rest of the subsamplers where filled much faster. This would,



when scaling the 3.75 % and the 10-minute exposure time, result in a cross-contamination of each sample with ~0.1 % of the previous one. Laboratory tests were carried out prior to the campaign, when adjacent loops of this exact subsampler were filled with clean air from the "working tank" (see section 2.3) and trace gas-free synthetic air. This yielded, after storage

220 times of up to 23 days, cross-port contamination effects of 0.2 – 0.5 %. Importantly, increasing storage time had no discernible effect on these numbers and there was also no detectable contamination with lab air. We cannot constrain this cross-contamination estimate further here, but it is worth keeping this effect in mind as something that could slightly bias the determined mole fractions, especially in regions of steep concentration gradients. However, as opposed to the first one, the other two subsamplers were newly built at the time of the campaign and are likely to have performed better due to having

experienced less wear and tear on their central valves. It is also important to note that cross-port contamination effects are greatly enhanced by pressure gradients between individual loops. This is something that cannot be easily simulated during the lab tests as evacuation of the inlet line (including the adjacent loop) needs to take place when switching between different filling gases. Such potential issues are therefore minimised with the applied constant pressure filling method. We conclude that the best estimate of cross-port contamination during the MAC subsampling is ~0.1 % or less except for the one outlier

discussed above.

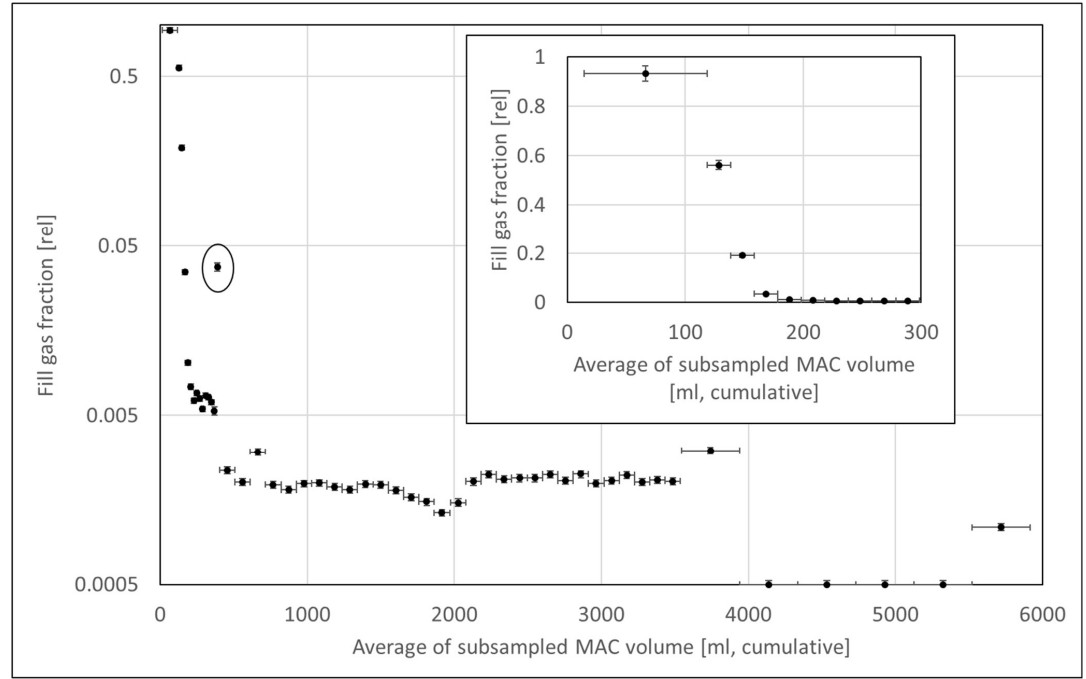



Figure 2: Cumulated fill gas fraction contained in the subsamples collected from the MAC as a function of the cumulative subsampled volume (averaged per sample) as determined from the content of PFTEA. The inset illustrates the rapid drop in fill gas fraction over the first 10 samples on a non-logarithmic scale. For four samples the PFTEA signal was too small to be quantified, these are shown as the detection limit (0.0005). Vertical error bars depict the analytically determined fraction uncertainty, whereas horizontal error bars refer to the sample size. The circled data point shows unusually high PFTEA, which is likely caused by the subsampling procedure. Further details can be found in Section 3.1.

## 3.2 Vertical atmospheric distributions of halogenated trace gases

Assigning altitudes to CRYO-based samples is rather straightforward as the times, pressures and GPS altitudes at the time of opening and closing the individual canisters are well known (Schuck et al., 2024). This procedure is more complicated for AirCores in general as only one sample is being collected over the entire flight. The altitude assignment procedure therefore ideally considers multiple influential factors such as pressure and temperature gradients inside the very long tubing, Taylor and molecular diffusion, or loss of air after landing due to the heating of the AirCore (Karion et al., 2010; Tans, 2022). This is somewhat simplified by the MAC as flown here, as the controlled and relatively slow descent speeds (Figure 3) prevent pressure gradients, and the active heating of the tubing does not generate any substantial temperature changes (see Section 2.2). In addition, the novel fill gas correction method allows for an improved and straightforward calculation of the portion of atmospheric air sampled. Therefore, the MAC altitudes can be derived by calculating the amount of air collected via a simple temperature and pressure correction (through the ideal gas law) of the fill gas-corrected number of moles inside each of the subsampler loops. This is done cumulatively starting with the lowest pressure reached during the flight, following the methodology described in Section S1.1.3 of Laube et al. (2020).

### 3.2.1 $SF_6$ and CFC-11

Figure 3 exemplary shows the vertical profiles of $SF_6$ and CFC-11 as derived from the MAC and CRYO. CRYO mole fractions of both species were merged from the measurements at GUF and FZJ and, for $SF_6$ only, primarily published in the companion paper of Schuck et al. (2024). For $SF_6$, very good agreement with the MAC is observed for 11 of the 13 CRYO samples. Especially notable is that both MAC and CRYO capture the relatively shallow layer of low $SF_6$ mixing ratios around 24 km that has been observed by $CH_4$ in situ measurements on the same gondola (Schuck et al., 2024). This is somewhat contrasting our previous estimate of the average molecular diffusion inside the MAC, which actually exceeds the subsample size at these altitudes (Section 2.2); although this might qualitatively be reconciled by the fill gas spiking correction being able to remove some of the mixing retrospectively (Section 3.1). The latter is especially pronounced at the high-altitude end of the profile where fill gas fractions are highest.



The two CRYO samples that for SF$_6$ do not agree with the MAC within their 1 sigma measurement uncertainties are close to
20 km, i.e., in the profile part with the steepest mixing ratio gradient. This discrepancy is likely caused by diffusion leading
to mixing inside the MAC, which is known to lead to smoothing effects in AirCores (e.g., Membrive et al., 2017). In this
case it results in an asymmetric shift in the profile towards higher MAC mole fractions due to the influence of relatively
stable mole fractions above ~24 km and further increasing ones below. This is exacerbated here by

i)        the relatively large internal diameter of the MAC not limiting diffusion as much as in the much narrower
commonly used lower-volume AirCores (Tans, 2022),
ii)        the long delay between the start of the balloon descent and the subsampling of the MAC (see Section 2.2), and
iii)        additional MAC smoothing being caused by the transition from 20 ml to ~100 ml subsamples which occurs at
~24 km, i.e., directly at the upper end of the largest mixing ratio gradient.

The MAC-CRYO comparison for CFC-11 supports the mixing effect hypothesis. Due to chemical decay in the stratosphere,
the mixing ratio gradient of this CFC with altitude is much more pronounced than for the nearly inert SF$_6$, which should lead
to more diffusion-induced mixing. This is exactly what is being observed, with five of the six CRYO samples collected
above 19 km exhibiting significantly lower mixing ratios than those of the MAC; the only exception being one of the two
CRYO samples at ~19.3 km, which was collected over a much larger altitude range than the adjacent one. Similar
differences are being observed for CH$_4$ when comparing CRYO samples and smaller AirCores in our companion paper
(Schuck et al., 2024).

Lastly for the MAC only, one sample at ~18.5 km looks anomalously low for SF$_6$, with no correlated value visible for CFC-
11 or almost all other species (except for C$_2$F$_6$ and CFC-13, both of which elute close in time to SF$_6$ during GC-MS
analysis). While we cannot fully rule out that there was a temporary measurement system glitch, we have kept these values
in the dataset as no obvious (physical) reason for excluding this sample has been found.





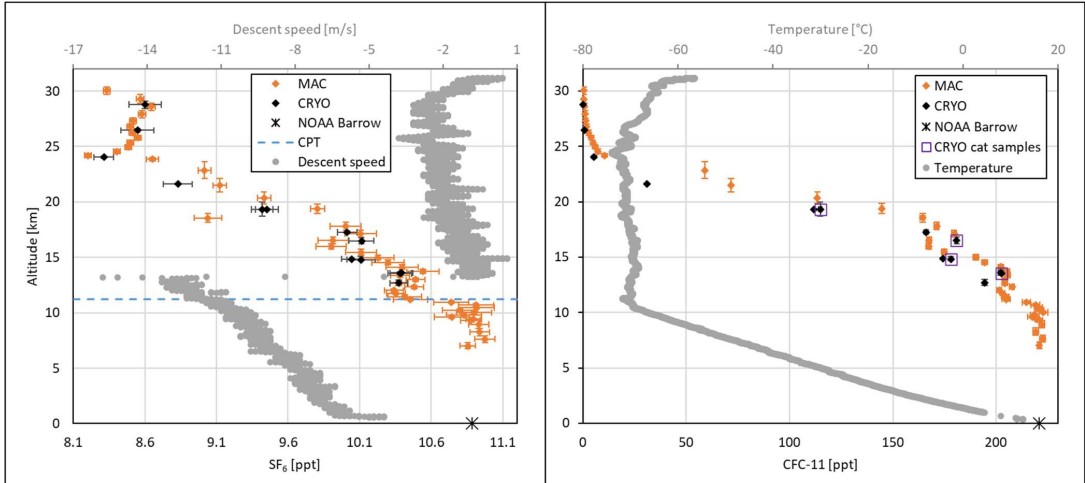

Figure 3: Left: Vertical distribution of $SF_6$ mixing ratios on 13[th] August 2021 near Kiruna, Sweden from the middle troposphere to the middle stratosphere as derived for the MegaAirCore (MAC, orange) and the Cryosampler (CRYO, black). Vertical error bars represent the altitude range over which was sampled, whereas horizontal error bars are equivalent to 1 standard deviation from the measurements (see text for further details). CRYO results are the average of multiple measurements carried out at the University of Frankfurt and Forschungszentrum Jülich. Also shown are i) the monthly mean mixing ratio of $SF_6$ from the polar ground-based observatory at Pt. Barrow, Alaska (black star, publicly available data from the NOAA Global Monitoring Labs: https://gml.noaa.gov), ii) the cold-point tropopause (CPT, blue-dashed), and iii) the descent speed of the gondola carrying the instrument (grey, on secondary x-axis). Three MAC samples with fill gas contents of 20 % or more have been excluded due to the additional uncertainties introduced by the fill gas content corrections (see Section 3.1). Right: The same as on the left, but for CFC-11, and with i) four CRYO samples with an $O_3$-scrubbing catalyst at the inlet highlighted (empty purple squares), and ii) ambient temperature plotted on the secondary x-axis (grey).

In addition, altitude-related discrepancies are observed when just looking at CRYO samples: Of the three sample pairs collected at similar altitudes, agreement within two standard deviations is observed for only one in the case of CFC-11. However, for $SF_6$ all three sample pairs agree within two standard deviations. This reflects a wider tendency as better agreement is found for longer-lived species such as CFC-115, HFC-23, or c-$C_4F_8$, which have less steep mole fraction gradients. Therefore, these differences are likely strongly influenced by the slightly different altitude ranges over which sample pairs were collected, in combination with strong mole fraction gradients in their immediate vicinity.





Also visible in Figure 3 is that the cut-off of the balloon with a much faster descent on a parachute from ~13.3 km
downwards has no apparent influence on the derived MAC mole fractions; whereas the cold-point tropopause at ~11.2 km
can be clearly identified as a transport barrier due to the disconnect between $SF_6$ and CFC-11 concentrations above and
below. Further down, it is reassuring to see that the middle tropospheric MAC data for both gases agree very well with the
ground-based mole fractions from a similar latitude, i.e., the observatory at Pt. Barrow, Alaska, USA (71.3N, 156.6W, part
of the NOAA Global Monitoring Labs global network).


### 3.2.2 Problematic species

The laboratory storage tests mentioned in Section 3.1 were also used to determine the long-term stability of the target species
prior to the campaign. No significant effects were found for most species after up to 23 days, although the spread of the
results from the individual subsampler loops was rather high at around 5 %. This is likely related to some shortcomings of
the testing method, with the uncertainties introduced by the fill-evacuate cycles as well as the slight differences in fill
pressure in each loop; none of which occur during MAC/AirCore subsampling. While this could mean that potentially all
mole fractions might be biased by up to 5 % this is highly unlikely due to

|      |      |
|------|------|
| i)   | the much more compact correlations observed in the MAC profile, |
| ii)  | the much-improved volume/internal surface area ratio in the larger subsamplers, and |
| iii) | the low mixing ratios observed for many gases near the top of the profile, which are often close to the detection limit (see, e.g., CFC-11 in Figure 3) and therefore evidencing the leak-tightness of the subsampling system. |

Storage stability exceptions were found only for some of the shorter-lived species ($CH_2Cl_2$: 1/3 loops +15 % after 23 days
only; $CHCl_3$: 2/3 loops at +10 and +16% after 23 days only; $CH_3Cl$: mixing ratios in all loops consistently increasing with
increasing storage time after more than 10 days by up to 30 %– consequently not measured in MAC) as well as HFC-23. The
latter is a well-known problem as small amounts of this gas are emitted by the rotor material (Valcon E®) in the central valve
of the subsamplers.


Accordingly, many trace gases show good comparability between CRYO and MAC (see H-1211 in Figure 4), but for some
there are issues with, e.g., contaminations (see Table S1). Figure 4 demonstrates this for CFC-12, for which two MAC
subsamplers are randomly contaminated leading to a break-down of its compact correlations with altitude and other trace
gases in the middle of the profile. The exact source of this contamination, which also affects c-$C_4F_8$, HFC-32, HFC-134a,
and HFC-227ea, could not be determined subsequently. It should however be noted that i) both subsamplers were newly
built for this flight and might have been insufficiently conditioned (i.e., flushed and heated) beforehand, and ii) this cannot



be a contamination with outside air since there were no leaks found and other substances were not affected at all. On the more positive side, the usable part of the data for HFC-32, HFC-143a, HFC-152a, HFC-236fa, and HFC-245fa are to our knowledge a unique in situ-based quantification of the stratospheric distribution of these species (disclaimer: only CRYO

GUF measurements available for the latter three of these HFCs).

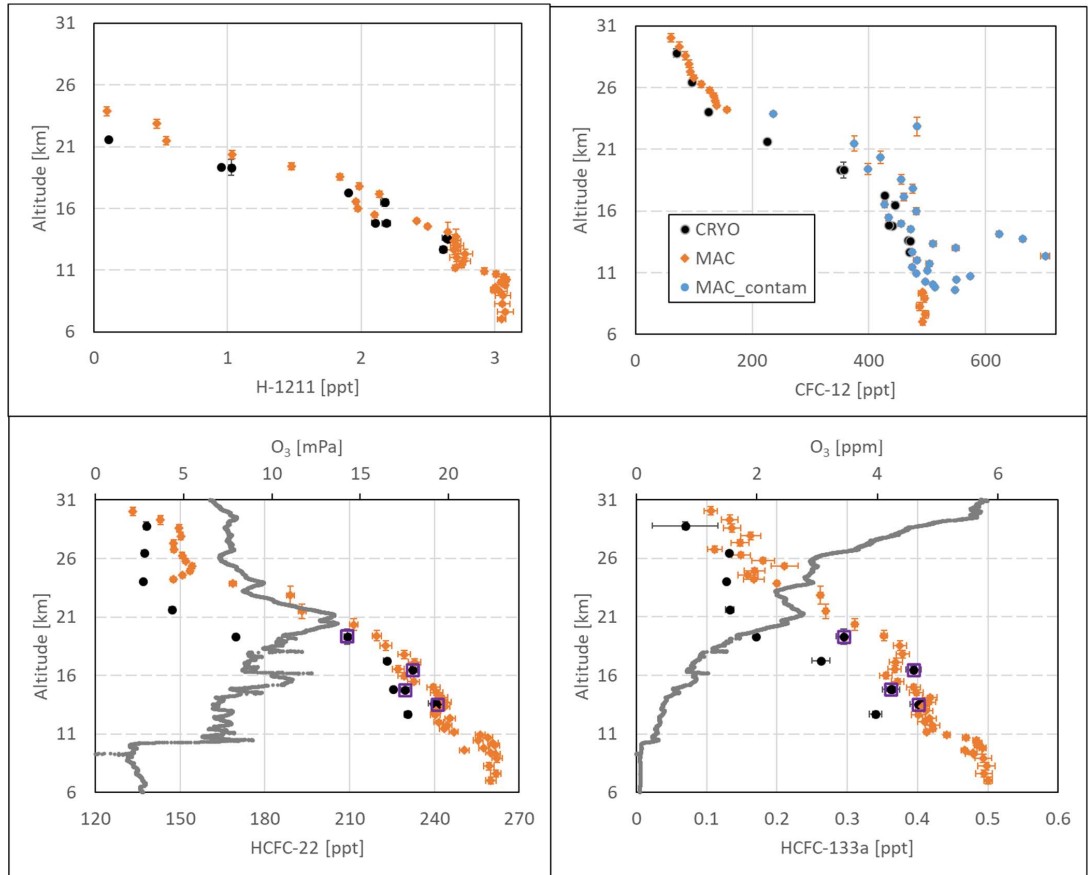

Figure 4. The same as in Figure 3 but for H-1211 (top left), CFC-12 (top right), HCFC-22 (lower left), and HCFC-133a (lower right), again with four CRYO samples with an $O_3$-scrubbing catalyst at the inlet highlighted (empty purple squares),

and with $O_3$ partial pressures and mixing ratios on secondary x axes (lower two panels).



Also shown in Figure 4 (lower two panels) are two HCFCs, i.e., HCFC-22 and HCFC-133a. Here – as well as for the other two HCFCs quantified (HCFC-141b and HCFC-142b) – larger differences are observed between MAC and CRYO samples. However, much better agreement is found for the four CRYO samples with the $O_3$ scrubbing catalyst in the inlet (highlighted in Figure 4). In the upper part of the profile, where $O_3$ mole fractions (on secondary x axis in Figure 4, derived from a concurrent ozone sonde launch) are still increasing, CRYO and MAC HCFC mole fractions converge again. While we cannot explain this convergence, this is overall a strong indication that the $O_3$ present in the CRYO samples might affect some species chemically during the sampling or storage process. Note that the main difference in sampling technique between the MAC and the CRYO is that the latter condenses the air and the trace gases which might allow for chemical processes that are otherwise not significant. Two further bits of evidence support this chemical decay hypothesis: Firstly, the vertical profiles derived from the MAC are very compact and follow the shape observed for other (more chemically inert) species such as $SF_6$ much more closely. Even the low mole fraction feature near 24 km is present in the MAC profile, but not for the CRYO. And secondly, the recent study of Kolonjari et al., (2024) compared stratospheric HCFC-22 CRYO data with those from a balloon-born remote-sensing instrument, two satellite instruments, and a model. They found that the CRYO samples often exhibited lower mole fractions (up to ~30 %), and this was especially pronounced in the upper part of their comparison altitude range (i.e., between 20 and 25 km). Nevertheless, we have no definitive proof that the CRYO samples are low-biased, nor do we have proof that the MAC samples are completely unbiased; although there are some strong pointers here, and more to follow in the next section.

### 3.2.3 Potential influence of air masses from the Asian Summer Monsoon

Figure 5 shows the vertical distributions of mole fractions for four species: $CCl_4$, CFC-113a, $CH_2Cl_2$, and $CHCl_3$. All of them are displaying a layer of increased MAC-based mole fractions between 9.4 and 10.9 km (potential temperature range: 300 to 305 K), i.e., just underneath the cold-point tropopause at 11.2 km. The mole fractions of $CH_2Cl_2$, and $CHCl_3$ show particularly high increases. Both gases are known to have increased in concentration in recent years with East Asia being a major driver of the emission increases (Fang et al., 2019; Claxton et al., 2020). Similarly, the Asian Summer Monsoon has been demonstrated as an effective transport mechanism for such emissions into the Upper Troposphere/Lower Stratosphere (UTLS) region, especially in the northern hemisphere (Vogel et al., 2019; Adcock et al., 2021; Lauther et al., 2022; Pan et al., 2024). Similarly, $CCl_4$ and CFC-113a are also known to have strong sources in the East Asian region (Adcock et al., 2018; Park et al., 2021). Following the methodology of a recent study by Graßl et al. (2024) (who demonstrated the impact of the Asian summer monsoon on the Arctic aerosol budget in summer 2021), we investigated whether artificial surface origin tracers released globally within the Chemical Lagrangian Model of the Stratosphere (CLaMS; e.g., Vogel et al., 2016) might be associated with the South Asian source region. The latter is defined here as in Graßl et al. (2024) and encompasses the monsoon region including India, Nepal, southern and eastern China, the Bay of Bengal and the Arabian Sea. Indeed, the model data shows some influence of this air (fractions from 1.5 to 3.5 %) in the vicinity of the launch site (Figures S3-S5),





although it appears to be at slightly higher altitudes (i.e., around 310 K). Note that the CLaMS output is always at 12 UTC, whereas the balloon flight took place during nighttime – and that the distribution of the tracer can change significantly within a few hours, especially in the UTLS. The observed mole fraction enhancement is relatively modest (~15 ppt for $CH_2Cl_2$, in comparison to several hundred ppt observed at these altitudes above East Asia). It is therefore plausible that the observed feature is indeed influenced by outflow from the Asian Summer Monsoon, which is also known to influence the extra-

tropical northern hemispheric UTLS the most in late summer and early autumn (e.g., Ploeger et al., 2015; Lauther et al., 2022).

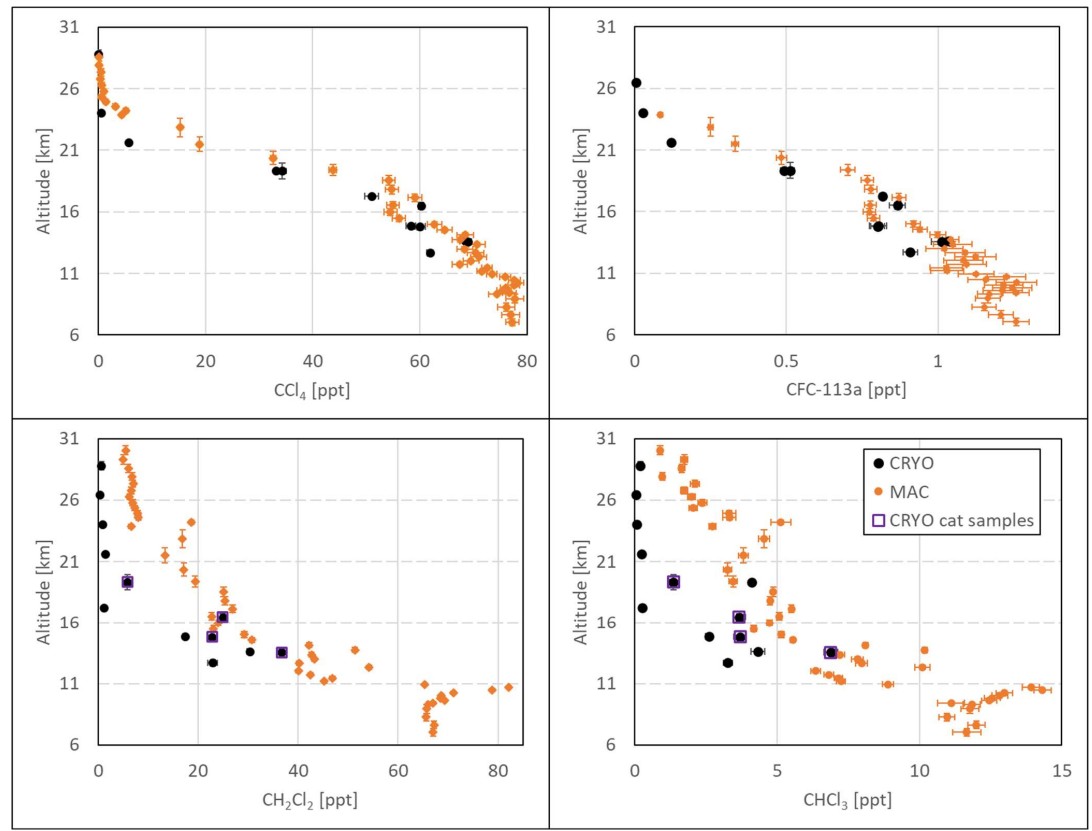

Figure 5. The same as in Figure 3 but for $CCl_4$ (top left), CFC-113a (top right), $CH_2Cl_2$ (lower left), and $CHCl_3$ (lower right).




Moreover, as the attentive reader has probably noticed in Figure 5, CRYO and MAC sample mole fractions are more scattered and do not agree very well with each other for $CH_2Cl_2$ and $CHCl_3$. Some of the enhanced MAC mole fractions coincide with samples contaminated with CFC-12 (see Section 3.2.2) but others do not. There might therefore be influence from the contamination of the middle two subsamplers here, especially recognisable for $CH_2Cl_2$, which exhibits a much more

compact correlation shape towards the top and bottom of the profile. However, the comparability of CRYO and MAC samples is not very good for most samples, and even catalyst-equipped ones appear not to perform consistently. As the internal surfaces of the CRYO canisters have no inert treatment, and long-term stability issues for more reactive compounds have been previously reported (Laube et al., 2008, Schuck et al., 2020), it is more likely that the uncontaminated MAC samples are the more realistic option here. Nevertheless, in the absence of further extensive stability testing under realistic

stratospheric conditions (including $O_3$), we recommend a cautious approach to the part of the data that consists of short-lived halogenated species.

### 3.2.4 Derivation of the mean age of air from multiple tracers

Data for three inert gases from the MAC ($SF_6$, $C_2F_6$, and HFC-125) and three further ones from the CRYO (HFC-23, $C_3F_8$,

and c-$C_4F_8$) is of sufficient quality to attempt the determination of the mean age of air (i.e., average stratospheric transit times, a key indicator for the overturning circulation strength). All these gases are principally suitable as age tracers as they fulfil the crucial criteria of chemical inertness as well as monotonically increasing mole fractions in the troposphere over at least a decade (see WMO, 2022 for recent trends and stratospheric lifetimes). For $SF_6$, the inertness criterion has been found to be not entirely true (Andrews et al., 2001; Leedham Elvidge et al., 2018), but we here employ a method based on a recent

study which allows for the correction of the related effects (Garny et al., 2024a). Also notable is that opposed to the other five gases, which have been known to be suitable for determining mean ages for some time (e.g., Volk et al., 1997; Andrews et al., 2001; Ray et al., 2017; Leedham Elvidge et al., 2018), c-$C_4F_8$ has only recently been introduced for this purpose by Umezawa et al. (2024). As for the method to derive the mean ages, we here employ a well-established parameterisation of the age spectrum assuming a constant value of 0.7 for the ratio of the squared width of the age spectrum with the mean age

(Leedham Elvidge et al., 2018). Following previous studies, we employ a well-known and robust method. The tropospheric long-term trends utilised are updated ones from Leedham Elvidge et al. (2018) for all gases except $SF_6$ (Ray et al., 2024) and c-$C_4F_8$ (Droste et al., 2020).

Figure 6 shows the mean ages as derived from all six gases. Note that the $SF_6$-based CRYO results are slightly different to

those reported in our companion paper (Schuck et al., 2024) as we employ a consistent mole fraction retrieval and uncertainty derivation method for both the FZJ and the GUF dataset. The related differences are well within those uncertainties though. In addition, $SF_6$-based ages have all been corrected for the bias introduced by the strong mesospheric sink of this molecule following the recently introduced method of Garny et al. (2024a). As expected from the mixing ratio



profile differences, the discrepancies between MAC and CRYO are on average greatest in the region with the steepest age
gradient, i.e., between 19 and 24 km. Otherwise the CRYO- and MAC-based mean ages generally agree well, though the
uncertainty ranges sometimes do not overlap. It should be noted, though, that the related error bars in Fig. 6 only represent
the measurement uncertainty of the stratospheric samples (1 standard deviation). Considering the uncertainties in the age
spectrum parameterisation as well as the tropospheric trend could add up to 1 month and up to about a year, respectively
(Leedham Elvidge et al., 2018; Umezawa et al., 2024). These are not included here to improve the clarity of Fig. 6.
However, when adding a) a 1-month age spectrum parameterisation uncertainty, and b) the tropospheric trend uncertainties
from Leedham Elvidge et al., 2018 for $C_2F_6$ and HFC-125, and the more recent ones from Umezawa et al. (2024) for all
other gases, all but three mean age estimates agree with those based on $SF_6$ (excluding non-physical negative mean ages near
or below the tropopause). Two of these are outliers observed for HFC-125 (Figure 6). Both can be found in the uppermost
subsampler and are most likely to be due to small contaminations, which in the case of age tracers are sufficient to induce a
substantial low bias to the derived mean ages.

Also apparent from Figure 6 is that the mean ages derived from $C_2F_6$ in the uppermost MAC subsampler are much more
scattered. This is likely due to the much smaller sample sizes (less than 1/5th) as compared to the samples below, which
limits the signal size for $C_2F_6$. The relevant error bars in Figure 6 therefore are likely an underestimate of the actual
uncertainty, highlighting the statistical limitations associated with very small sample sizes that do not allow for repeat
measurements. As for the CRYO-based estimates, the mean ages derived from $C_3F_8$ and c-$C_4F_8$ compare well with those
from the other tracers and have much smaller error bars than those reported in Umezawa et al. (2024). This is entirely due to
the better stratospheric measurement precisions (~1% as compared to ~6 %, see Table S1) in this work, and underlines the
principal suitability of these two species as age tracers.




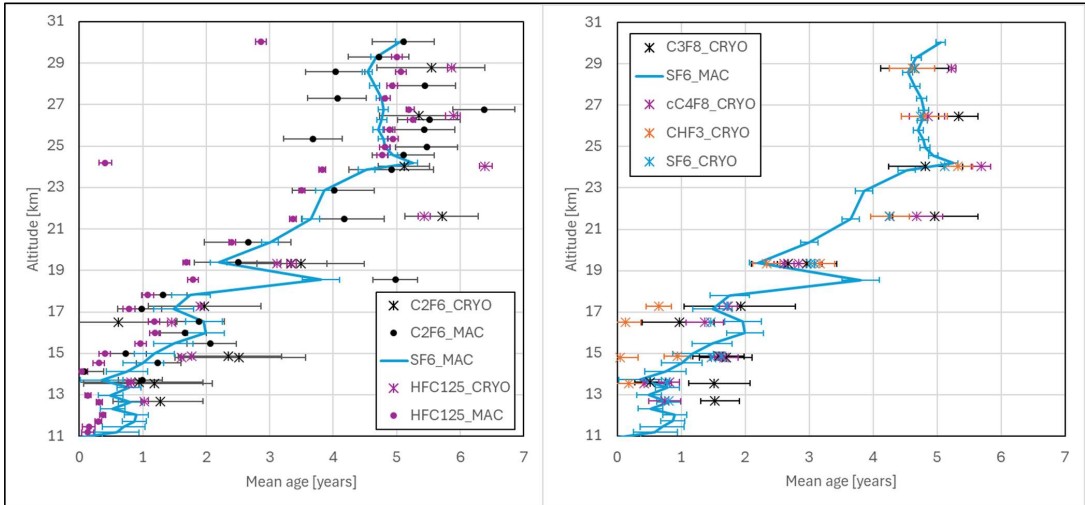

Figure 6. Left: Mean ages of air versus altitude as derived from tropospheric trends and stratospheric mole fraction as available from the MAC and the CRYO samplers for $C_2F_6$, $SF_6$ (MAC only), and HFC-125. Right: The same but for $C_3F_8$,
$SF_6$ (including CRYO), c-$C_4F_8$, and $CHF_3$. All $SF_6$-based ages have been corrected according to Garny et al. (2024a).

## 4 Conclusions

In this study we analysed subsampled air from a very large AirCore for its content of 30 halogenated trace gases and demonstrated the useability of 26 of those (Table S1). As the dataset shows, close attention needs to be paid to the leak-tightness of the entire AirCore and subsampling system as well as thorough flushing to prevent as many contaminations as
possible. A new method to correct for the remaining fill gas in the MAC by spiking it with a gas a) measurable with mass spectrometry alongside the target gases, and b) currently not present in the atmosphere, was employed successfully. In addition, the CRYO enabled access to even more trace species due to the larger sample size but was also found to exhibit previously unreported problems with some species (HCFCs) that were thought to be stable in this sampler. Here, the catalytic destruction of $O_3$ in the sample canister inlet led to a much better agreement with the MAC-based samples, pointing
towards chemical decay of some species during cryosampling and/or storage. Overall, the quality of the MAC data is good for a maiden flight, with several of the more inert species yielding sensible and CRYO-comparable results when deriving mean ages.

Generally, the MegaAirCore (MAC) profiles successfully demonstrate the advantages of a large AirCore compared to a
cryogenic whole-air sampler (CRYO): 48 instead of 13 viable air samples could be obtained, revealing multiple layering



structures in various parts of the stratosphere (similar to the ones observed by the collocated in situ $CH_4$ instrument in Schuck et al. (2024)). This indicates a particularly high vertical stability, which is probably at least in part due to the very limited stratospheric temperature gradient (Figure 3) and the timing of the campaign close to the stratospheric wind turnaround in August. In addition, the MAC enables access to multiple samples deep inside the troposphere, thus allowing

for an easier comparison to ground-based measurements. This is something that is not always achievable with the CRYO as the due to the limited number of samples the collection is often focused on the harder-to-access higher altitudes. However, cryogenically collected samples allow for larger volumes and are more suitable for collecting air samples over very narrow and well-defined altitude ranges. This could be improved for AirCores by better limiting internal mixing (which can also introduce biases in the vertical distribution, especially in the vicinity of strong concentration gradients), e.g., through further

minimising the delay between landing and subsampling – although the feasibility here is very much dependent on the accessibility of the terrain in the landing area. Smaller AirCores generally offer easier and therefore faster recoveries, albeit at the cost of less air being available for trace gas analysis. Similarly, lowering the size of the subsamples taken from AirCores can help to better capture smaller features and/or strong gradients in the vertical mole fraction distributions (e.g., Li et al., 2023) as is demonstrated in the upper part of the MAC profile.


Importantly, this is not a full verification of the AirCore sampling technique as the large balloon utilised here descended much slower for much of the flight. Pressure non-equilibrium effects from the near-freefall at high altitudes during common AirCore flights remain unverified in situ, as do the additional temperature-induced changes in the very cold stratosphere when flying an unheated AirCore. Overall, this dataset adds substantially to the rather sparse in situ-based mean age record

(Garny et al., 2024b), and could in the future be, e.g., useful for a) validating satellite products (Saunders et al., 2024), and b) deriving constraints on the shape of age spectra with new approaches (Voet et al., 2024; Ray et al., 2024) through the diversity of trace gases that are available.

*Data availability.* Observational data will be made available in a public domain repository after acceptance of this
manuscript.

*Author contributions.* JCL, TJS, HC, SvH TK, MEP, TW, AZ, and AE contributed to campaign preparation (including laboratory work on the design and testing of the $O_3$ scrubbers) and the operation of the instrumentation in the field. TJS and JCL performed post-flight sample analysis. MG and ET contributed to data analysis and interpretation, whereas BV provided
CLaMS model output. JCL drafted the manuscript, and all coauthors contributed to improving it.

*Competing interests.* At least one of the authors is a member of the editorial board of Atmospheric Measurement Techniques.



*Acknowledgements*. We are extremely grateful for the support from the technical staff before, during, and after the campaign,
especially from Anne Richter, Andreas Sitnikov, Jochen Barthel, and Vicheith Tan (FZJ), Laurin Merkel (GUF), and Carina
van der Veen (Utrecht). Also invaluable was the support from teams at ESRANGE base, from CNES as flight operator and
provider of additional ozone sonde data, and from Mélanie Ghysels, Georges Durry, and Nadir Amarouche (provision of
ambient pressure data). We acknowledge the work of the NOAA Global Monitoring Laboratory for providing surface
measurements of multiple gases for comparison with our data, as well as for the derivation of age of air in the case of $SF_6$.

*Financial support*. This research has been supported by the European Research Council (grants no. 678904 (EXC3ITE) and
no. 742798 (COS-OCS; to M. C. Krol)), by the DFG collaborative research program "The Tropopause Region in a Changing
Atmosphere" (TRR 301 – Project-ID 428312742), the EU INFRAIA grant 730790-HEMERA, and the Forschungszentrum
Juelich GmbH.

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
