# Peer review of "Vertical distribution of halogenated trace gases in the summer Arctic stratosphere based on two independent air sampling methods"

_EGUsphere, 2024_

## Author Response (AR1)

**Note to the editor and editorial team**

Please be aware that we have added two extra coauthors, i.e. Sophie Baartman (who contributed to the cryosampler preparations including catalyst design and testing) as well as Florian Voet (who contributed to the improved calculations of age of air). In addition, altitudes assigned to the MAC have slightly changed in the revised version due to a small improvement in the algorithm – but all changes are very minor, i.e. < 9 meters. Lastly, we have removed the plural of "AirCore" or changed it to "AirCore devices" following a similar request by the editorial team during the review process of our companion paper (Schuck et al., 2025).

**Responses to reviewer comments**

RC1: 'Comment on egusphere-2024-4034', Anonymous Referee #1, 12 Feb 2025

Review of "Vertical distribution of halogenated trace gases in the summer Arctic stratosphere determined by two independent in situ methods" by Laube et al.

General comment

This manuscript presents measurement results from two types of balloon-borne air samplers to study vertical distribution of halogenated trace gases in the stratosphere. The authors combined conventional and newly developed air-sampling and analytical techniques which required very careful and complex steps. Although there seem to be issues for some species, I would like to send congratulation to the authors for the reasonable success. The manuscript is well structured and written. I recommend publication of this manuscript after the minor comments below are considered.

Author response: We thank the reviewer for the positive feedback as well as the constructive comments!

Reviewer comment

In section 2.3, the authors described possible diffusion effects that deteriorate vertical resolution of the measurements. Since the diffusion is molecular dependent, it means that altitudinal resolution is different from species to species, as the authors explains with Figures S1 and S2. I wonder if the curves in these figures could be interpreted as uncertainties of the altitude assignment of the measurement data and if this was considered in presented figures. The authors also infer that the diffusion would more affected $CO_2$ and $CH_4$. As the measurement data of these species have been presented in the companion paper (Shuck et al. 2025) (and more commonly in previous AirCore measurements), it might be worth presenting magnitudes of the diffusion effect in $CO_2$ and $CH_4$ as well.

Author response:

The curves referred to here do influence altitude uncertainties but represent the smoothing of the vertical distributions rather than altitude uncertainty. The full altitude uncertainty also depends on multiple other factors like the uncertainties of the fill gas mixing correction, the volume determination of the subsamplers, the exact pressure in each subsample, and the precision of the pressure and temperature measurements during both flight and subsampling. Nevertheless, the reviewer is correct that molecular diffusion is the main component of the altitude uncertainty. We have therefore added the maximum molecular diffusion as outer error bars in Fig. 3 and added the following explanation to the caption:

"Vertical error bars represent the altitude range over which was sampled, whereas, for the MAC only, vertical outer (dashed) error bars represent the altitude range equivalent to root mean square molecular diffusion."

..as well as adapting the discussion in Section 3.2.1:

"Nevertheless, we have also displayed outer error bars in Figure 3, which represent the altitude range equivalent to root mean square molecular diffusion (see Fig. S1 for more calculation details). It should be noted, though, that these are not actual altitude uncertainties but represent the altitude range over which diffusion has an influence."

"ii) the long delay between the start of the balloon descent and the subsampling of the MAC (see Section 2.2) *leading to more molecular diffusion (as indicated by the outer error bars in Fig. 3)*"

In addition, we have added molecular diffusion for CO2 and CH4 to Fig. S1 as suggested, but merged all four vertical profiles into one figure:

"To provide visual comparability, similar molecular diffusion plots for $CO_2$ and $CH_4$ have also been included in the supplement (Fig. S1)."

Reviewer comment

The section also presents the interesting and decisive component of this study, subsampling. Because I have found information in the reference (Laube et al. 2020) is limited, I understand that this paragraph (P2 L118ff) will be reference for the authors' future measurements. Although I likely followed their system and steps, it might have be much easier and clearer for readers and future studies if a schematic figure of the subsampling system is presented.

Author response:

We appreciate the suggestion and have added a schematic to the supplement, which is also linked to in that paragraph:

"(see schematic in Figure S2)."

Reviewer comment

As written in the manuscript (Figure 3 caption), presented for the CRYO samples are from the averages of measurements at both FZJ and GUF. I would consider that presentation of the data from both labs in the figures might be another option. It would also allow profile comparison between the CRYO and MAC data from the single lab (FZJ), which could highlight difference from sampling techniques only. In addition, some general explanation about agreement between the data from both labs, at least for representative species in central discussion in this study, might be added. One option could be to move the supplement text (P1) to Section 2.4. Table S2 needs more reader-friendly sentences in its caption (see the comment below).

Author response:

We are concerned that the separation of these two data sets in the plots might impede the visual clarity. In addition, since the measurements agree very well with each other, the reader might not be able to spot most differences. We are also hesitant to move all of the quality assurance details into the main manuscript as we wish the focus to remain on the more inventive aspects. However, we agree that comparability of measurements should be quantitatively mentioned in the main manuscript and have modified the statement at the end of the paragraph to incorporate this:

"As a brief summary of this comparison, 50 % of measurements agreed within one standard deviation, and 45 % within two standard deviations. More details can be found in the supplement."

Moreover, we have added the individual data sets to the merged one as part of the data published alongside the manuscript in an online repository: https://doi.org/10.5281/zenodo.15437807.

Specific comment

Reviewer comment

Title: I am not sure if these are called "in situ" methods. The methods are not certainly remote sensing, and air samples were collected in situ in the stratosphere, but the methods include subsequent lab analyses. I am confused at least with "determined by…in situ methods".

Author response:

We agree and have changed the title to "Vertical distribution of halogenated trace gases in the summer Arctic stratosphere based on two independent air sampling methods".

Reviewer comment

P3 L68: "the low weight" it would be good to mention to the exact weight here

Author response: We have added "(typically between 2 and 5 kg)".

Reviewer comment

P3 L73: "withing" to "within"

Author response: Well spotted, thank you! This has been corrected.

Reviewer comment

P12 L337: For CFC-12, the figure shows the data deemed as contaminated in blue, while the valid data are in orange. How did the classification work? There are the data with clear excursions from an expected profile. On the other hand, there are also those apparently aligned near the CRYO data but considered as contaminated.

Author response:

We have added an explanation of the classification to the text: "All data from those two subsamplers were excluded for a given trace gas if clear contaminations (defined as variability outside three standard deviation measurement uncertainties) were observed for more than two samples."

Reviewer comment

Table S2: needs more explanations in the caption. What e.g., y.y.y means should be explained in the caption in a full sentence (not as part of the table). What the colors indicate? What are number columns at right? Now readers have to read between the lines.

Author response:

We agree and apologise for this oversight. The caption has been extended as follow:

"Listed in each cell are agreement within one, two, and three standard deviations, with "n" referring to "no" and "y" to "yes". Colouring also refers to agreement within one (white), two (yellow), three (orange), or more (red) standard deviations. Blue fields indicate that the species was not detected by one or both measurement systems. The five columns on the righthand side represent the sum of data agreement cases for each species and have the same colour coding as the main part of the table."

RC2: 'Comment on egusphere-2024-4034', Anonymous Referee #2, 25 Mar 2025

Reviewer comment

This study investigates two different methods for sampling the atmospheric chemical composition from the mid troposphere to the mid stratosphere up to 30 km. The high altitude balloon borne cryogenic whole air sampler for stratospheric sampling has been used multiple times in the past. The AirCore sampler has been used less frequently for stratospheric sampling and, as such, required additional evaluation. This manuscript provides a detailed evaluation of the sampling and analytical methods used for both samplers. The manuscript is very well written and is clear in the detailed evaluations of the sampler performances. Given the difficulty in sampling the stratosphere, this work is exceptionally important, especially with the higher vertical resolution of the AirCore and the high cost of cryogen for the cryogenic sampler. As a next step, comparison of the cryogenic sampler and the AirCore with an in situ instrument that measures a subset of the organic halogens would be extremely informative. I commend the authors for their excellent treatment of the comparison of the two samplers and I recommend publication of this manuscript after addressing the minor comments/questions below.

Author response: We thank the reviewer for this comprehensive and positive assessment of our work!

Reviewer comment

Why were cotton filters used to remove ozone, e.g. have they been used in the past or evaluated relative to other potential methods? Were they tested in the lab to determine their ozone removal efficiency?

Author response: Cotton filters are an established method to remove O3 from air samples without affecting other trace gases. We have added two references to demonstrate that:

Hofmann, U., Hofmann, R., and Kesselmeier, J.: Cryogenic trapping of reduced sulfur compounds using a nafion drier and cotton wadding as an oxidant scavenger, Atmos. Environ. A-Gen., 26, 2445–2449, https://doi.org/10.1016/0960-1686(92)90374-T, 1992.

Persson, C. and Leck, C.: Determination of reduced sulfur compounds in the atmosphere using a cotton scrubber for oxidant removal and gas chromatography with flame photometric detection, Anal. Chem., 66, 983–987, https://doi.org/10.1021/ac00079a009, 1994.

In addition, the cotton filters were tested in the lab under various conditions and found to perform excellently for CO2, CH4, and OCS. More details will be published in a forthcoming manuscript. We take the point from the reviewer that currently there is virtually no information on those filters in the manuscript and have added the following: "These filters are known to remove O3 from air samples without affecting other trace gases (Hofmann et al., 1992; Persson et al., 1994). They were tested

prior to the campaign under various conditions and found to perform excellently for CO2, CH4, and OCS. A manuscript with more lab test details is in preparation."

Reviewer comment

Were the interior of the cryogenic sampling canisters coated with anything to improve trace gas inertness?

Author response: Alas, this was not possible and goes some way to explain the instabilities found for more reactive species, which have already been noted in previous studies (e.g., Laube et al., 2008, which is already cited for that very reason). The cryosampler cans are however electro-polished, which has been added to Section 2.1.

Reviewer comment

It's not clear to me how molecular diffusion is taken into account when evaluating mixing in the tubes and how mixing is taken into account when determining sample altitude. Perhaps an equation in the supplemental material would be helpful.

Author response: Indeed, we had not explained our method – thank you! The text of Section 3.2 now points the reader in the right direction ("Nevertheless, we have also displayed outer error bars in Figure 3, which represent the altitude range equivalent to root mean square molecular diffusion (see Fig. S1 for more calculation details)"), plus we have expanded the caption of Fig. S1 as follows:

"This was calculated as in Tans, 2022: $X_{rms} = \sqrt{2Dt}$ (t: time of air inside MAC) with diffusivity D from Eq. 2 in Kouznetsov et al., 2020: $D = D_0 \frac{p_0}{p} \left(\frac{T}{T_0}\right)^{3/2}$, and molecular diffusivities $D_0$ from Martinerie et al., 2009."

Reviewer comment

Table S2, what do the colors mean and what do the n's and y's mean?

Author response:

We apologise for this oversight. The caption has been extended as follow:

"Listed in each cell are agreement within one, two, and three standard deviations, with "n" referring to "no" and "y" to "yes". Colouring also refers to agreement within one (white), two (yellow), three (orange), or more (red) standard deviations. Blue fields indicate that the species was not detected by one or both measurement systems. The five columns on the righthand side represent the sum of

data agreement cases for each species and have the same colour coding as the main part of the table."

Reviewer comment

An interesting feature in the altitude profiles is the near constant mixing ratios between about 11 and 14 km rather than decreasing rapidly just above the cpt. I realize the focus of the manuscript is on the comparisons, but it might be worth a mention of what you think is responsible for this part of the profiles.

Author response:

Thank you for this observation and suggestion! The likeliest explanation for this gradual decrease is that there is mixing between air masses that are a) predominantly tropospheric in origin and have entered the stratosphere in the extra-tropics (e.g. through isentropic processes), and b) air masses that have been in the stratosphere for longer and have initially entered it through the tropical tropopause layer. This is a well-known process that is most pronounced in the extra-tropical tropopause region in late summer and autumn (e.g., Hauck et al., ACP, 2020). The hypothesis is supported by a) the more gradual increase of O3 mixing ratios in that altitude region (lower right in Figure 4), and b) the altitude discrepancy between the cold-point tropopause at 11.2 km and the tropopause as defined via WMO (i.e., where the temperature change with altitude first reaches the value of 2K km-1), which is at 10.5 km (Schuck et al., 2025). Such differences are indicative of a weak transport barrier between troposphere and stratosphere. We have added a short statement to section 3.2.1:

"Notable is also the region between the tropopause and about 14 km, which shows a less pronounced gradient in trace gas mole fraction than further up. This is likely a region influenced by enhanced troposphere-stratosphere exchange, which is often found in the extra-tropical tropopause region in late summer and autumn (e.g., Hauck et al., 2020). Such a hypothesis is supported by the observed discrepancy between the CPT at 11.2 km and the WMO-defined tropopause at 10.5 km (Schuck et al., 2025), which indicates a weakened tropopause transport barrier."

Reviewer comment

Regarding the use of a constant value of 0.7 for the squared width of the age spectrum with mean age, it has, as the authors point out, been used in the past. However, according to Ray, et al., 2024 "Most studies of age of air have used values of $R$ ranging from 0.7-1.25 years based on model estimates (e.g., Hall and Plumb, 1994; Volk et al., 1997; Engel et al., 2008). However, with a better understanding of the effect of the exponential tail of $G$ for $t' > 10$ years, the model estimates of $R$ have increased to values of 1.5 years or more with considerable variability in the stratosphere (e.g., Diallo et al., 2012; Ploeger and Birner, 2016; Fritsch et al., 2020)." I'm not suggesting the authors change the value they use, but they might consider qualifying their use beyond it's been used in the past.

Author response: This criticism is well justified, and we have therefore updated all our mean age estimates using state-of-the-art methodologies throughout, i.e. not only those published in Garny et al. 2024a (SF6 sink correction), but also in Garny et al., 2024b (new consolidated reference method for age of air calculations, used as described in the corresponding software publication of Wagenhäuser et al., 2024). This takes care of the expressed concerns on the consideration of the tail of the age spectrum as well as delivering a more refined representation of the ratio of the squared width of the age spectrum with mean age. As the resulting changes were small, the discussion of the results remains unaffected, so only the introductory text of Section 3.2.4, as well as Figure 6 have been updated accordingly:

"As for the method to derive the mean ages, we here employ a new consolidated reference method for parameterisation of the age spectrum Garny et al., 2024b; corresponding software in Wagenhäuser et al., 2024). This robust and state-of-the-art method includes a good representation of a) the long tail of the age spectrum as well as b) the squared width of the age spectrum with mean age."

RC3: 'Comment on egusphere-2024-4034', Anonymous Referee #3, 29 Mar 2025

Reviewer comment

This paper describes the results of a large stratospheric balloon flight in August 2021, including a large air-core sampler for halogenated compounds and a cryogenic whole air sampler, along with other instruments. Results from the two methods for halogenated compounds are compared and the relative strengths of each are described; results from the other instruments are described in a companion paper.

This is an important advance in stratospheric sampling and well suited for publication in Atmospheric Measurement Techniques. It is to my knowledge the first time that AirCores have been flown on a large balloon payload together with established instruments for halogenated compounds that sample air at reasonably well-defined altitudes. The large number of compounds measured and quantified is also quite impressive. The results of the flight were generally very good, and some important lessons were learned and discussed concerning both techniques and how to carry out this type of experiment. Many of the details of the balloon launch itself are in the companion paper by Schuck et al.; this is fine but hopefully the two papers will appear close in time (or even simultaneously, though in this day of electronic publishing the idea of "back-to-back" publications is perhaps a thing of the past).

Author response: We thank the reviewer for this comprehensive and positive assessment of our manuscript!

Reviewer comment

The techniques used in this balloon flight are generally described well, the figures are clear, and the results are novel. In particular, the use of the "MegaAirCore" with extensive subsampling is an important step forward and resulted in a large number of samples in the stratosphere, at altitudes up to 30 km with excellent vertical resolution. My only comment here is that previous work is barely mentioned. Laube et al. 2020 used AirCore results together with aircraft measurements and models to probe changes in the stratospheric distributions of halogenated compounds, and Li et al., 2023 described a technique to measure some of the same molecules directly from an AirCore into a gas chromatograph. Are there any other relevant publications on halogenated molecules in AirCores? How does this new publication build on previous work (much of it by the same author)? This may only need a few sentences or a short paragraph to address in the Introduction, or Section 2.2; no need to make this manuscript much longer.

Author response:

Alas, to the best of our knowledge there are no published AirCore-based measurements of halogenated compounds other than the two that are already referenced. We have also refrained from adding to the manuscript to elucidate further on how this work builds on previous publications, as we feel that the current statement in the Introduction offers a focused and relatively concise assessment of the heritage and main aims of this work:

"Here we present a unique dataset based on the established AirCore passive sampling technique (Karion et al., 2010; Membrive et al., 2017; Engel et al., 2017; Laube et al., 2020; Wagenhäuser et al., 2021; Tans, 2022). In this case, a very large AirCore was flown alongside a cryogenic whole-air sampler, which allows for an independent verification of the AirCore method; something that has not been performed for halogenated species before. A similar verification for the major anthropogenic greenhouse gases $CO_2$, $CH_4$, $N_2O$, and $SF_6$ can be found in our companion paper, i.e., Schuck et al. (2024). In addition, due to the increased volume of the stratospheric and upper tropospheric air that was retrieved with this AirCore, a much-improved number of halogenated trace species (as compared to the previously published seven in Laube et al., 2020, and Li et al., 2023) could be quantified (including CFCs, halons, HCFCs, HFCs, PFCs, and chlorocarbons), alongside retaining a relatively good altitude resolution. Describing the details of the employed methods and evaluating and comparing the results from both instruments are the primary focus of this manuscript."

Reviewer comment

I also found Figure 3 (one of the most important figures in the manuscript) and the text surrounding it a little confusing. First, the legend and caption refer to the CRYO samples with the O3 scrubber as open squares, but the squares on the right panel are not completely open, and seem to have something inside (in contrast to the legend). Or are there regular (non-scrubbed) data points inside as well? This seems to be the case in Figures 4 and 5 as well. This is also relevant to the statement

about agreement of CFC-11 on line 302. However, the CFC-11 CRYO data points at similar altitudes are very close to each other, and the explanation on lines 306-307 seems entirely reasonable. The low mixing ratios of SF6 near 24 km in both instruments are very interesting. I don't see how the (relatively small at this altitude) correction for residual fill gas could allow the recovery of this structure; perhaps the mixing did not occur completely or is less efficient than expected. This also calls into question the explanation for the disagreement between the two methods near 20 km, which would be caused by mixing of the air collected at 24 km with very low SF6 (lower than the cryo-sample) and air with higher SF6 collected at lower altitudes. The points i and iii starting at line 270 about the larger diameter of the AirCore and the larger sample volume are well-taken, however. The gradient in CFC-11 from 19-27 km is more readily explained in terms of mixing, since the MAC gradient is less steep than for the CRYO data. But for SF6, the gradient in the MAC data from 19-24 km is actually steeper than for CRYO data (if you believe the minimum at ~24 km, which is observed by both techniques). This may not be a terribly important point; both data sets look very interesting and the comparison and combination of the two have led/will lead to additional insights into both techniques as well as possible changes in how to plan for and conduct balloon flights.

Author response:

We thank the reviewer for these observant comments and the critical assessment! As for the symbols depicting catalyst samples in Figure 3, we have clarified their meaning in the caption as follows: "(black circles surrounded by empty purple squares)".

We also agree with the analysis of the unusual feature at 24 km and believe that especially the larger sample volume plays a dominant role for the discrepancies between MAC and CRYO. We have therefore added the following statement to reflect this: "This is likely the dominant factor here, as otherwise the capturing of the sharp feature of low SF6 mixing ratios at 24 km cannot be explained."

Reviewer comment

P.2, line 41 – Are there six or seven gases measured in AirCores in Laube et al., 2020? And there are really only five halogenated gases measured in Li et al., 2023 (six gases in all, but one of them is N2O).

Author response:

The additional halogenated trace gas covered in Li et al. is CFC-113. We have added a list of all seven species (SF6, CFC-11, CFC-12, CFC-113, HCFC-22, H-1211, and H-1301) after the references to make this clearer.

Reviewer comment

l.49 Schuck et al., 2024 also contains a few more details about the balloon flight, etc.; the authors could add something like ", along with additional details about the balloon flight" at the end, or something like that. (For example, I was curious how long the balloon stayed at altitude, and that is found in Schuck et al. No need to repeat it here.)

Author response:

Thank you for the good idea! We have added the suggested statement as is.

Reviewer comment

P.3, l. 78-95, The MegaAirCore is very interesting. Can it be flown by itself on smaller balloons?

Author response:

We certainly agree! It could in principle be flown by itself on smaller balloons, although at the typical upper end of the more affordable "small balloon" range are 3000g balloons. Flying the MAC on such a balloon would be possible, but limit the maximum achievable altitude to around 22 km. This would not be optimal as the remaining fill gas fraction inside the MAC would be much larger, thus adding considerably to the derived uncertainties; while at the same time not playing to one of the main strengths of such balloons, i.e. the achievable altitude range. We have however added the weight of the MAC to Section 2.2 to aid the reader's assessment of such possibilities: "The MAC weighs around 20 kg and was constructed specifically for this flight."

Reviewer comment

P.6, l. 179 I thought the addition of the perfluoro amine compound was an interesting (and seemingly very helpful) innovation, along with figuring out how to account for its possible changing detector response (on the following page). As long as it is not "sticky" in any of the tubing or valves, etc. it should work fine. It certainly was useful in pointing out the possible leakage in the last loop of the first subsampler (P.7, l. 211-215), and then (I think) correcting for it.

Author response:

We thank the referee for this assessment, with which we very much agree. The fact that the PFTEA-based calculation of the amount of fill gas remaining in the MAC at top altitude agrees with the pressure-based one is a strong argument for PFTEA being indeed not sticky.

Reviewer comment

P.8, l. 218-221 – Was there a pressure gradient between the two adjacent loops in this test? If not, it seems different than conditions that may have led to the outlier sample described on the previous page. Were these results used to correct the trace gas concentrations in that sample? And in Figure 1, P. 8, does the apparent fill gas fraction change abruptly on the logarithmic scale from sampler 1 to sampler 2? In any case, the explanations all seem reasonable.

Author response:

There always is a period of pressure gradient as loops are being filled. During those tests the pressure gradient was present during the evacuation period before filling the adjacent loop with a different gas. To aid the reader we have modified the respective statement: "It is also important to note that cross-port contamination effects are greatly enhanced by pressure gradients between individual loops. This is something that occurs during the lab tests as evacuation of the inlet line (including the adjacent loop) needs to take place when switching between different filling gases. Such potential issues are however minimised with the applied constant pressure filling method."

We have also complemented the initial statement on the MAC outlier to make clear that there was a pressure gradient: "The likeliest explanation for this outlier is therefore some cross-port leakage during that time, aided by the ~1 bar pressure gradient between the evacuated loop and the MAC."

As for the other questions: 1) These test results were not used to correct the trace gas concentrations in that sample and such a claim is not made in the paper. 2) There is indeed a step change in the fill gas fraction at the transition from sample 1 to sampler 2. This can be explained by the change in sample loop volume from ~20 to ~100 ml. The sudden averaging over a much larger volume in which the fill gas fraction decreases would be expected to lead to this effect.

Reviewer comment

P.9, l. 245-252 It seems that it might be worthwhile to calculate the altitudes in the MAC using the Tans method, if only for comparison.

Author response:

This would be an interesting addition, but, alas, also a rather work intensive one. The Tans method was designed for a different purpose, i.e., to derive altitudes from continuous analyser-based data. In addition, and as pointed out in the Conclusions, the MAC flight was not a very typical AirCore flight as it experiences much slower descend speeds, much larger internal diameters and included active heating of the tubing. We therefore refrain from adapting the Tans method for our needs here, also as we believe this to not add substantially to the core messages of this manuscript.

Reviewer comment

P.11, l. 303 "agreement within two standard deviations", compared to the Figure 3 caption, lines 292-293 "horizontal error bars are equivalent to 1 standard deviation". These may both be true, but that doesn't seem to be the clearest way of communicating this information. I agree with the point on l. 306 about "slightly different altitude ranges"; for a compound like CFC-11 with a large gradient and excellent measurement precision small (real) atmospheric variations are quite possibly the cause of some apparent disagreement.

Author response:

We have modified the first statement to improve clarity: "..agreement within even two standard deviations (instead of the one standard deviation in Fig. 3) is observed for only one in the case of CFC-11".

We also agree with the idea on potential atmospheric variability as a factor, and have modified the statement to reflect this: "Therefore, these differences are likely strongly influenced by the slightly different altitude ranges over which sample pairs were collected, in combination with pronounced mole fraction gradients in their immediate vicinity as well as, potentially, atmospheric variability."

Reviewer comment

P.12, l. 330-331 – I really don't understand what "1/3 loops" means. Is it that in one third of the loops, the mixing ratio increased by 15%. And why "only"? And in the next line "+10 and +16%"?

Author response:

We have modified the first example to help make this clearer: "($CH_2Cl_2$: 1 out of 3 loops +15 % only in the case of maximum storage time, i.e., after 23 days". This should also explain the comment on the next line as the "+10 and +16%" refer to the 2 (out of 3) loops.

Reviewer comment

l. 337-338 The contamination for CFC-12 does not look random at all. It only appears in the middle section of the data (~10km-23km). Or do you mean that within one (or two) subsamplers the contamination is random, and some of the cyan points are actually not contaminated?

Author response:

We have modified the sentence to make clearer that the middle two subsamplers are randomly contaminated: "Figure 4 demonstrates this for CFC-12, for which the middle two MAC subsamplers are randomly contaminated leading to a break-down of its compact correlations with altitude and other trace gases in the middle of the profile."

Reviewer comment

P.15, l. 386-387 Can the distribution and mixing ratios of the surface origin tracer be diagnosed in the CLaMS model on a faster than daily timescale? If model output is once a day (at 12 UTC), how can one know that it changes "significantly within a few hours"? Perhaps I am mixing up model and measurements (or knowledge from past aircraft campaigns); but if so then the text could be made clearer.

Author response:

Different setups of the model are able to focus on shorter time-scale variability. We have added a statement and reference to elucidate this, including a justification for not using them here: "Past studies have used more sophisticated techniques to trace the air mass origins (e.g., Adcock et al., 2021). This is however beyond the scope of this technique-focused study."

Reviewer comment

P.16, l. 414 Shouldn't the Ray et al., 2017 paper on the lifetime of SF6 go here? I am not sure if this is the correct spot for Andrews et al., 2001 either (I did not go back and look at that one), though it should certainly be included on line 416 (as it is).

Author response:

Andrews et al. is correct here, but Ray et al., 2017 also fits very well and has therefore been added.

Reviewer comment

l. 419-420, For this methods paper, using a constant value of 0.7 is fine, though it surely varies, at least somewhat. And the sentence "Following previous studies…" seems redundant and unnecessary.

Author response:

The entire age of air calculation has been updated to a recently published consolidated reference method (Garny et al., 20224b) – for details please see our response to a similar comment by RC2.

Reviewer comment

P.17, l. 433 Uncertainties in the SF6 tropospheric trend do not add a year of uncertainty; perhaps for other trace gases used they can. I would say that the parameterization of the age spectrum could add at least a month; if it were only a month that would be great.

Author response:

We agree. This is why we state that it can add "up to about a year", including a reference, both for this claim as well as the one on the age spectrum parameterisation.

Reviewer comment

P.19, l. 472 - Is the vertical stability really particularly high? Is it different than other similar high latitude profiles in summer, or is this time/region typically very stable? If temperature actually increased with altitude, the atmospheric stability would be even stronger.

Author response:

We agree and have modified the statement to: "This indicates a particularly high vertical stability, which is probably at least in part due to the generally limited vertical mixing/exchange processes in the middle stratosphere that lead to the long transport timescales of the residual overturning circulation."

Technical and proofreading comments:

P.3, l. 66 This sentence could just start with "AirCores were invented [or developed] at the NOAA…"

P.6, l. 177-178 – You might consider putting the word "exceptions" in the earlier sentence to avoid somewhat contradicting yourself. Something like "…trace gas-free N2, with two exceptions. These were: i) small …"

P.9, l. 233 – It should probably be "Cumulative" instead of "cumulated", but more importantly, the fill gas fraction is not cumulated. Only the subsampled volume is cumulative. At least that is how the figure appears to me.

l. 254 "Figure 3 shows the vertical profiles…" is good enough.

P.10, l. 279-280 "Similar differences were observed for CH4 when comparing CRYO samples and smaller AirCores (Schuck et al., 2024)."

P.19, l. 476 "as due to"

Author response:

Thank you for spotting those, all of which have been adjusted as requested.